# Deconstructing Lottery Tickets:
# Zeros, Signs, and the Supermask

**Hattie Zhou**
Uber
hattie@uber.com

**Janice Lan**
Uber AI
janlan@uber.com

**Rosanne Liu**
Uber AI
rosanne@uber.com

**Jason Yosinski**
Uber AI
yosinski@uber.com

## Abstract

The recent "Lottery Ticket Hypothesis" paper by Frankle & Carbin showed that a simple approach to creating sparse networks (keeping the large weights) results in models that are trainable from scratch, but only when starting from the same initial weights. The performance of these networks often exceeds the performance of the non-sparse base model, but for reasons that were not well understood. In this paper we study the three critical components of the Lottery Ticket (LT) algorithm, showing that each may be varied significantly without impacting the overall results. Ablating these factors leads to new insights for why LT networks perform as well as they do. We show why setting weights to zero is important, how signs are all you need to make the reinitialized network train, and why masking behaves like training. Finally, we discover the existence of Supermasks, masks that can be applied to an untrained, randomly initialized network to produce a model with performance far better than chance (86% on MNIST, 41% on CIFAR-10).

## 1 Introduction

Many neural networks are over-parameterized [3, 4], enabling compression of each layer [4, 21, 8] or of the entire network [14]. Some compression approaches enable more efficient computation by pruning parameters, by factorizing matrices, or via other tricks [8, 10, 13, 16–18, 20–23]. Unfortunately, although sparse networks created via pruning often work well, training sparse networks directly often fails, with the resulting networks underperforming their dense counterparts [16, 8].

A recent work by Frankle & Carbin [5] was thus surprising to many researchers when it presented a simple algorithm for finding sparse subnetworks within larger networks that *are* trainable from scratch. Their approach to finding these sparse, performant networks is as follows: after training a network, set all weights smaller than some threshold to zero, pruning them (similarly to other pruning approaches [9, 8, 15]), rewind the rest of the weights to their initial configuration, and then retrain the network from this starting configuration but with the zero weights frozen (not trained). Using this approach, they obtained two intriguing results.

First, they showed that the pruned networks performed well. Aggressively pruned networks (with 95 percent to 99.5 percent of weights pruned) showed no drop in performance compared to the much larger, unpruned network. Moreover, networks only moderately pruned (with 50 percent to 90 percent of weights pruned) often outperformed their unpruned counterparts. Second, they showed that these pruned networks train well only if they are rewound to their initial state, including the specific initial weights that were used. Reinitializing the same network topology with new weights causes it to train poorly. As pointed out in [5], it appears that the specific combination of pruning mask and weights underlying the mask form a more efficient subnetwork found within the larger network, or, as named by the original study, a lucky winning "Lottery Ticket," or LT.

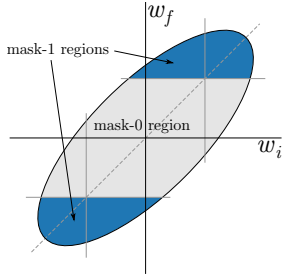

Figure 1: Different mask criteria can be thought of as segmenting the 2D ($w_i$ = initial weight value, $w_f$ = final weight value) space into regions corresponding to mask values of 1 vs 0. The ellipse represents in cartoon form the area occupied by the positively correlated initial and final weights from a given layer. The mask criterion shown, identified by two horizontal lines that separate the whole region into mask-1 (blue) areas and mask-0 (grey) areas, corresponds to the large_final criterion used in [5]: weights with large final magnitude are kept and weights with final values near zero are pruned.

While Frankle & Carbin [5] clearly demonstrated LT networks to be effective, it raises many intriguing questions about the underlying mechanics of these subnetworks. What about LT networks causes them to show better performance? Why are the mask and the initial set of weights so tightly coupled, such that re-initializing the network makes it less trainable? Why does simply selecting large weights constitute an effective criterion for choosing a mask? We attempt to answer these questions by exploiting the essential steps in the lottery ticket algorithm, described below:

0. Initialize a mask $m$ to all ones. Randomly initialize the parameters $w$ of a network $f(x; w \odot m)$

1. Train the parameters $w$ of the network $f(x; w \odot m)$ to completion. Denote the initial weights before training $w_i$ and the final weights after training $w_f$.

2. *Mask Criterion.* Use the mask criterion $M(w_i, w_f)$ to produce a masking score for each currently unmasked weight. Rank the weights in each layer by their scores, set the mask value for the top $p\%$ to 1, the bottom $(100 - p)\%$ to 0, breaking ties randomly. Here $p$ may vary by layer, and we follow the ratios chosen in [5], summarized in Table S1. In [5] the mask selected weights with large final value corresponding to $M(w_i, w_f) = |w_f|$.

3. *Mask-1 Action.* Take some action with the weights with mask value 1. In [5] these weights were reset to their initial values and marked for training in the next round.

4. *Mask-0 Action.* Take some action with the weights with mask value 0. In [5] these weights were pruned: set to 0 and frozen during any subsequent training.

5. Repeat from 1 if performing iterative pruning.

In this paper we perform ablation studies along the above three dimensions of variability, considering alternate mask criteria (Section 2), alternate mask-1 actions (Section 3), and alternate mask-0 actions (Section 4). These studies in aggregate reveal new insights for why lottery ticket networks work as they do. Along the way we discover the existence of Supermasks—masks that produce above-chance performance when applied to untrained networks (Section 5). We make our code available at `https://github.com/uber-research/deconstructing-lottery-tickets`.

## 2 Mask criteria

We begin our investigation with a study of different *Mask Criteria*, or functions that decide which weights to keep vs. prune. In this paper, we define the mask for each individual weight as a function of the weight's values both at initialization and after training: $M(w_i, w_f)$. We can visualize this function as a set of decision boundaries in a 2D space as shown in Figure 1. In [5], the mask criterion simply keeps weights with large final magnitude; we refer to this as the large_final mask, $M(w_i, w_f) = |w_f|$.

We experiment with mask criteria based on final weights (large_final and small_final), initial weights (large_init and small_init), a combination of the two (large_init_large_final and small_init_small_final), and how much weights move (magnitude_increase and movement). We also include random as a control case, which chooses masks randomly. These nine masks are depicted along with their associated equations in Figure 2. Note that the main difference between magnitude_increase and movement is that those weights that change sign are more likely to be kept in the movement criterion than the magnitude_increase criterion.

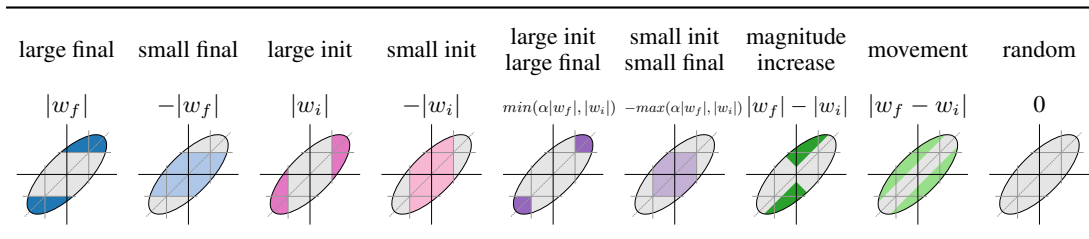

| large final | small final | large init | small init | large init large final | small init small final | magnitude increase | movement | random |
|---|---|---|---|---|---|---|---|---|
| $\|w_f\|$ | $-\|w_f\|$ | $\|w_i\|$ | $-\|w_i\|$ | $min(\alpha\|w_f\|,\|w_i\|)$ | $-max(\alpha\|w_f\|,\|w_i\|)$ | $\|w_f\|-\|w_i\|$ | $\|w_f-w_i\|$ | $0$ |

Figure 2: Mask criteria studied in this section, starting with large_final that was used in [5]. Names we use to refer to the various methods are given along with the formula that projects each $(w_i, w_f)$ pair to a score. Weights with the largest scores (colored regions) are kept, and weights with the smallest scores (gray regions) are pruned. The $x$ axis in each small figure is $w_i$ and the $y$ axis is $w_f$. In two methods, $\alpha$ is adjusted as needed to align percentiles between $w_i$ and $w_f$. When masks are created, ties are broken randomly, so a score of 0 for every weight results in random masks.

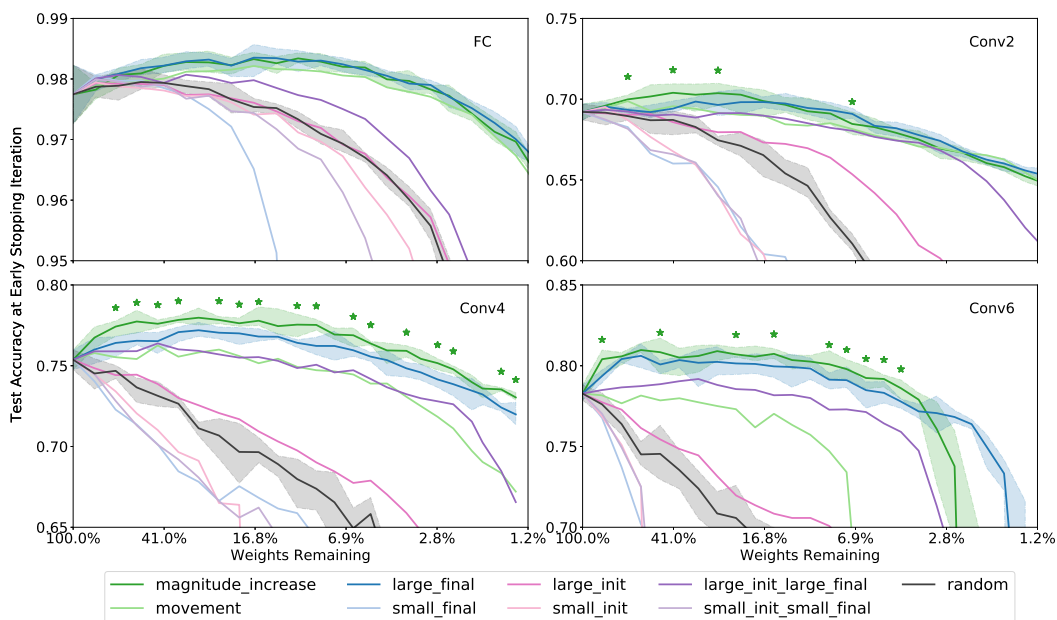

Figure 3: Test accuracy at early stopping iteration of different mask criteria for four networks at various pruning rates. Each line is a different mask criteria, with bands around the best-performing mask criteria (large_final and magnitude_increase) and the baseline (random) depicting the min and max over 5 runs. Stars represent points where large_final or magnitude_increase are significantly above the other at $p < 0.05$. The eight mask criteria form four groups of inverted pairs (each column of the legend represents one such pair) that act as controls for each other. We observe that large_final and magnitude_increase have the best performance, with magnitude_increase having slightly higher accuracy in Conv2 and Conv4. See Figure S1 for results on convergence speed.

In this section and throughout the remainder of the paper, we follow the experimental framework from [5] and perform iterative pruning experiments on a 3-layer fully-connected network (FC) trained on MNIST [12] and on three convolutional neural networks (CNNs), Conv2, Conv4, and Conv6 (small CNNs with 2/4/6 convolutional layers, same as used in [5]) trained on CIFAR-10 [11]. For more architecture and training details, see Section S1 in Supplementary Information. We hope to expand these experiments to larger datasets and deeper models in future work. In particular, [6] shows that the original LT Algorithm as proposed do not generalize to ResNet on ImageNet. It would be valuable to see how well the experiments in this paper generalize to harder problems.

Results of all criteria are shown in Figure 3 for the four networks (FC, Conv2, Conv4, Conv6). The accuracy shown is the test accuracy at an early stopping iteration[1] of training. For all figures in this paper, the line depicts the mean over five runs, and the band (if shown) depicts the min and max obtained over five runs. In some cases the band is omitted for visual clarity.

Note that the first six criteria as depicted in Figure 2 form three opposing pairs; in each case, we observe when one member of the pair performs better than the random baseline, the opposing member performs worse than it. Moreover, the magnitude_increase criterion turns out to work just as well as the large_final criterion, and in some cases significantly better[2].

The conclusion so far is that although large_final is a very competitive mask criterion, the LT behavior is not limited to this mask criterion as other mask criteria (magnitude_increase, large_init_large_final, movement) can also match or exceed the performance of the original network. This partially answers our question about the efficacy of different mask criteria. Still unanswered: why either of the two front-running criteria (magnitude_increase, large_final) should work well in the first place. We uncover those details in the following two sections.

## 3 Mask-1 actions: the sign-ificance of initial weights

Now that we have explored various ways of choosing which weights to keep and prune, we will consider how we should initialize the kept weights. In particular, we want to explore an interesting observation in [5] which showed that the pruned, skeletal LT networks train well when you rewind to its original initialization, but degrades in performance when you randomly reinitialize the network.

Why does reinitialization cause LT networks to train poorly? Which components of the original initialization are important? To investigate, we keep all other treatments the same as [5] and perform a number of variants in the treatment of 1-masked, trainable weights, in terms of how to reinitialize them before the subnetwork training:

- "Reinit" experiments: reinitialize kept weights based on the original init distribution.
- "Reshuffle" experiments: reinitialize while respecting the original distribution of remaining weights in that layer by reshuffling the kept weights' initial values.
- "Constant" experiments: reinitialize by setting 1-masked weight values to a positive or negative constant; thus every weight on a layer becomes one of three values: $-\alpha$, $0$, or $\alpha$, with $\alpha$ being the standard deviation of each layer's original initialization.

All of the reinitialization experiments are based on the same original networks and use the large_final mask criterion with iterative pruning. We include the original LT network ("rewind, large final") and the randomly pruned network ("random") as baselines for comparison.

We find that none of these three variants alone are able to train as well as the original LT network, shown as dashed lines in Figure 4. However, all three variants work better when we ensure that the new values of the kept weights are of the same sign as their original initial values. These are shown as solid color lines in Figure 4. Clearly, the common factor in all working variants including the original rewind action is the sign. As long as you keep the sign, reinitialization is not a deal breaker; in fact, even setting all kept weights to a constant value consistently performs well! The significance of the sign suggests, in contrast to [5], that the basin of attraction for an LT network is actually quite large: optimizers work well anywhere in the correct sign quadrant for the weights, but encounter difficulty crossing the zero barrier between signs.

## 4 Mask-0 actions: masking is training

What should we do with weights that are pruned? This question may seem trivial, as deleting them (equivalently: setting them to zero) is the standard practice. The term "pruning" implies the dropping of connections by setting weights to zero, and these weights are thought of as unimportant. However, if the value of zero for the pruned weights is not important to the performance of the network, we should expect that we can set pruned weights to some other value, such as leaving them frozen at

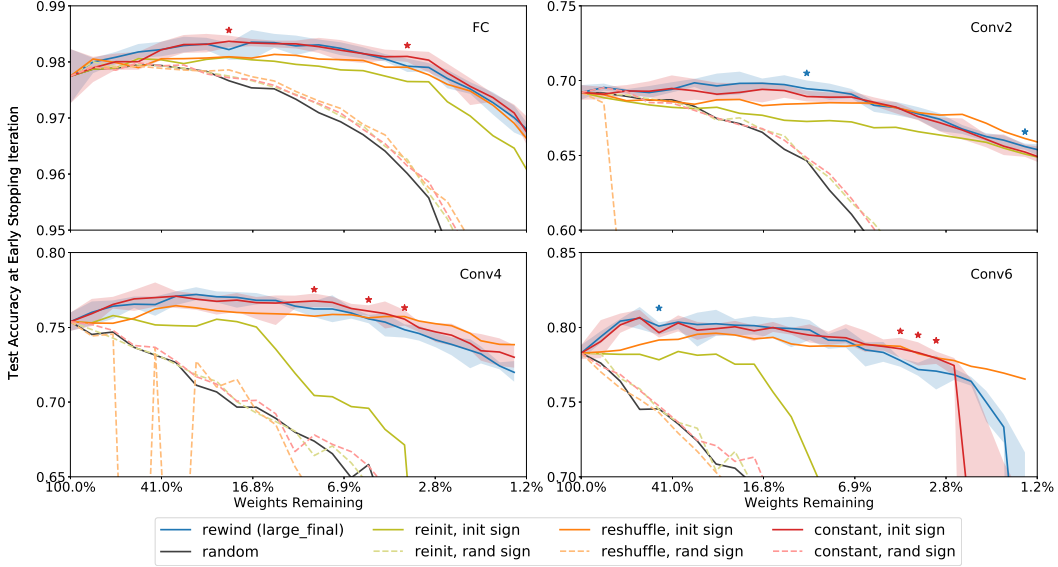

Figure 4: The effects of various 1-actions for four networks at various pruning rates. All reinitialization experiments use the `large_final` mask criterion with iterative pruning. Dotted lines represent the three described methods, and solid lines are those three except with each weight having the same sign as its original initialization. Shaded bands around notable runs depict the min and max over 5 runs. Stars represent points where "rewind (large_final)" or "constant, init sign" is significantly above the other at a $p < 0.05$ level, showing no difference in performance between the two. The original `large_final` and `random` are included as baselines. See Figure S4 for results on convergence speed.

their initial values, without hurting the trainability of the network. This turns out to not be the case. We show in this section that zero values actually matter, alternative freezing approach results in better performing networks, and masking can be viewed as a way of training.

Typical network pruning procedures [9, 8, 15] perform two actions on pruned weights: set them to zero, and freeze them in subsequent training (equivalent to removing those connections from the network). It is unclear which of these two components leads to the increased performance in LT networks. To separate the two factors, we run a simple experiment: we reproduce the LT iterative pruning experiments in which network weights are masked out in alternating train/mask/rewind cycles, but try an additional treatment: freeze masked weights at their initial values instead of at zero. If zero isn't special, both should perform similarly.

Figure 5 shows the results for this experiment. We find that networks perform significantly better when weights are frozen specifically at zero than at random initial values. For these networks masked via the LT `large_final` criterion[3], zero would seem to be a particularly good value to set pruned weights to. At high levels of pruning, freezing at the initial values may perform better, which makes sense since having a large number of zeros means having lots of dead connections.

So why does zero work better than initial values? One hypothesis is that the mask criterion we use *tends to mask to zero those weights that were headed toward zero anyway.*

To test out this hypothesis, we propose another mask-0 action halfway between freezing at zero and freezing at initialization: for any zero-masked weight, freeze it to zero if it moves toward zero over the course of training, and freeze it at its random initial value if it moves away from zero. We show two variants of this experiment in Figure 5. In the first variant, we apply it directly as stated to zero-masked weights (to be pruned). We see that by doing so we achieve comparable performance to the original LT networks at low pruning rates and better at high pruning rates. In the second variant, we extend this action to one-masked weights too, that is, initialize every weight to zero if they move towards zero during training, regardless of the pruning action on them. We see that performance of Variant 2 is even better than Variant 1, suggesting that this new mask-0 action we found can be

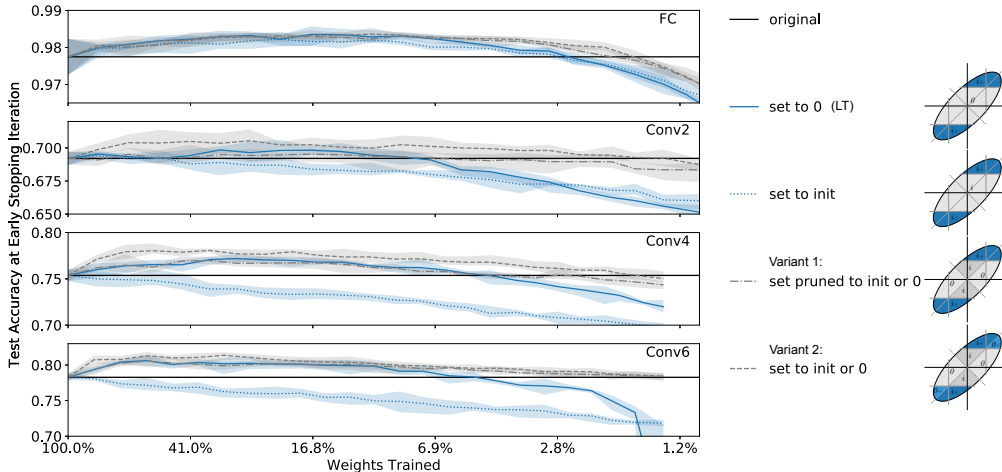

Figure 5: Performance of network pruning using different treatments of pruned weights (mask-0 actions). Horizontal black lines represent the performance of training the original, full network, averaged over five runs. Solid blue lines represent the original LT algorithm, which freezes pruned weights at zero. Dotted blue lines freeze pruned weights at their initial values. Grey lines show the new proposed 0-action—set to zero if they decreased in magnitude by the end of training, otherwise set to their initialization values. Two variants are shown: 1) new treatment applied to only pruned weights (dashdotted grey lines); 2) new treatment applied to all weights (dashed grey lines).

a beneficial mask-1 action too. These results support our hypothesis that the benefit derived from freezing values to zero comes from the fact that those values were moving toward zero anyway[4]. This view on masking as training provides a new perspective on 1) why certain mask criteria work well (large_final and magnitude_increase both bias towards setting pruned weights close to their final values in the previous round of training), 2) the important contribution of the value of pruned weights to the overall performance of pruned networks, and 3) the benefit of setting these select weights to zero as a better initialization for the network.

# 5   Supermasks

The hypothesis above suggests that for certain mask criteria, like large_final, that masking is training: *the masking operation tends to move weights in the direction they would have moved during training*. If so, just how powerful is this training operation? To answer this, we can start from the beginning— not training the network at all, but simply applying a mask to the randomly initialized network.

It turns out that with a well-chosen mask, an untrained network can already attain a test accuracy far better than chance. This might come as a surprise, because if you use a randomly initialized and untrained network to, say, classify images of handwritten digits from the MNIST dataset, you would expect accuracy to be no better than chance (about 10%). But now imagine you multiply the network weights by a mask containing only zeros and ones. In this instance, weights are either unchanged or deleted entirely, but the resulting network now achieves nearly 40 percent accuracy at the task! This is strange, but it is exactly what we observe with masks created using the large_final criterion.

In randomly-initialized networks with large_final masks, it is not implausible to have better-than-chance performance since the masks are derived from the training process. The large improvement in performance is still surprising, however, since the only transmission of information from the training back to the initial network is via a zero-one mask based on a simple criterion. We call masks that can produce better-than-chance accuracy without training of the underlying weights "Supermasks".

We now turn our attention to finding better Supermasks. First, we simply gather all masks instantiated in the process of creating the networks shown in Figure 2, apply them to the original, randomly initialized networks, and evaluate the accuracy without training the network. Next, compelled by the demonstration in Section 3 of the importance of signs and in Section 4 of keeping large

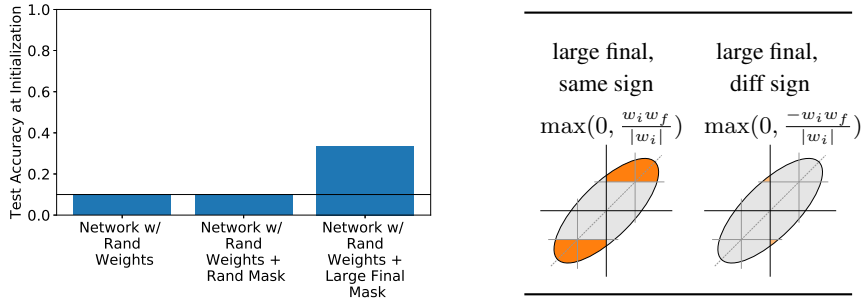

Figure 6: **(left)** Untrained networks perform at chance (10% accuracy) on MNIST, if they are randomly initialized, or randomly initialized and randomly masked. However, applying the large_final mask improves the network accuracy beyond the chance level. **(right)** The large_final_same_sign mask criterion (left) that tends to produce the best Supermasks. In contrast to the large_final mask in Figure 1, this criterion masks out the quadrants where the sign of $w_i$ and $w_f$ differ. We include large_final_diff_sign (right) as a control.

weights, we define a new large_final_same_sign mask criterion that selects for weights with large final magnitudes that also maintained the same sign by the end of training. This criterion, as well as the control case of large_final_diff_sign, is depicted in Figure 6. Performances of Supermasks produced by all 10 criteria are included in Figure 7, compared with two baselines: networks untrained and unmasked (untrained_baseline) and networks fully trained (trained_baseline). For simplicity, we evaluate Supermasks based on one-shot pruning rather than iterative pruning.

We see that large_final_same_sign significantly outperforms the other mask criteria in terms of accuracy at initialization. We can create networks that obtain a remarkable 80% test accuracy on MNIST and 24% on CIFAR-10 without training using this simple mask criterion. Another curious observation is that if we apply the mask to a signed constant (as described in Section 3) rather than the actual initial weights, we can produce even higher test accuracy of up to 86% on MNIST and 41% on CIFAR-10! Detailed results across network architectures, pruning percentages, and these two treatments, are shown in Figure 7.

We find it fascinating that these Supermasks exist and can be found via such simple criteria. As an aside, they also present a method for network compression, since we only need to save a binary mask and a single random seed to reconstruct the full weights of the network.

## 5.1 Optimizing the Supermask

We have shown that Supermasks derived using simple heuristics greatly enhance the performance of the underlying network immediately, with no training involved. In this section we are interested in how far we can push the performance of Supermasks by *training the mask*, instead of training network weights. Similar works in this domain include training networks with binary weights [1, 2], or training masks to adapt a base network to multiple tasks [19]. Our work differs in that the base network is randomly initialized, never trained, and masks are optimized for the original task.

We do so by creating a trainable mask variable for each layer while freezing all original parameters for that layer at their random initialization values. For an original weight tensor $w$ and a mask tensor $m$ of the same shape, we have as the effective weight $w' = w_i \odot g(m)$, where $w_i$ denotes the initial values weights are frozen at, $\odot$ is element-wise multiplication and $g$ is a point-wise function that transform a matrix of continuous values into binary values.

We train the masks with $g(m) = \text{Bern}(S(m))$, where $\text{Bern}(p)$ is the bernoulli sampler with probability $p$, and $S(m)$ is the sigmoid function. The bernoulli sampling adds some stochasticity that helps with training, mitigates the bias of all things starting at the same value, and uses in effect the expected value of $S(m)$, which is especially useful when they are close to 0.5.

By training the $m$ matrix with SGD, we obtained up to 95.3% test accuracy on MNIST and 65.4% on CIFAR-10. Results are shown in Figure 7, along with all the heuristic-based, unlearned Supermasks. Note that there is no straightforward way to control for the pruning percentage. Instead, we initialize $m$ with larger or smaller magnitudes, which nudges the network toward pruning more or less. This

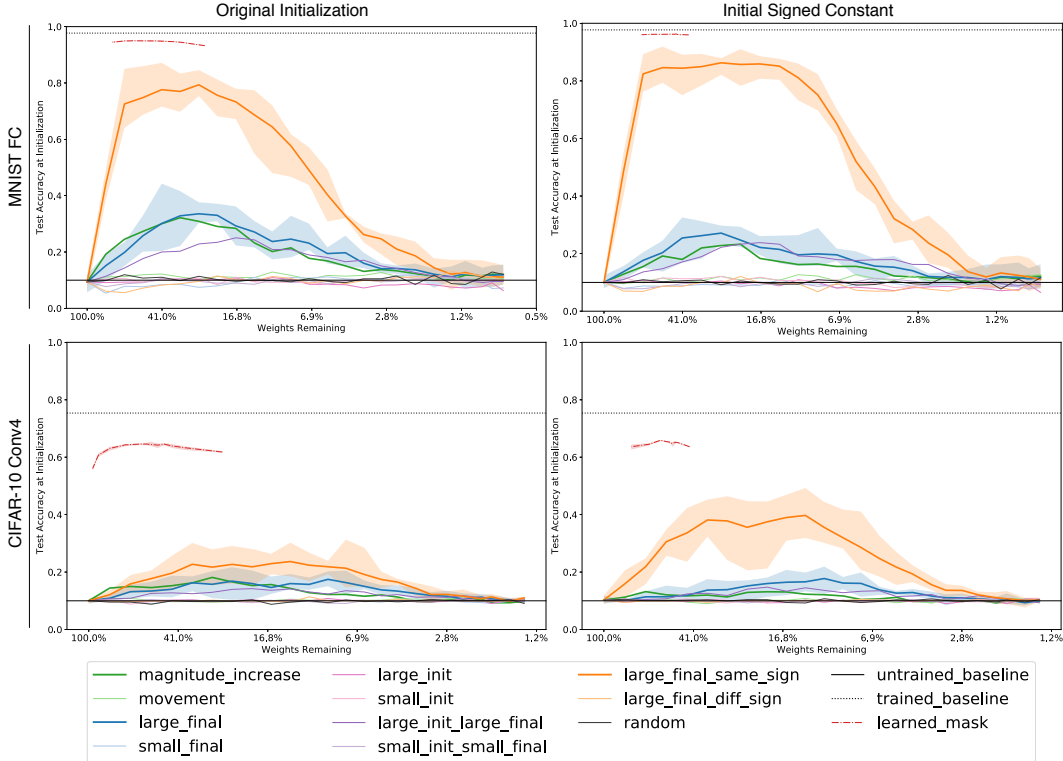

Figure 7: Comparision of Supermask performances in terms of test accuracy on MNIST and CIFAR-10 classification tasks. Subfigures are across two network structures (top: FC on MNIST, bottom: Conv4 on CIFAR-10), as well as 1-action treatments (left: weights are at their original initialization, right: weights are converted to signed constants). No training is performed in any network. Within heuristic based Supermasks (excluding learned_mask), the large_final_same_sign mask creates the highest performing Supermask by a wide margin. Note that aside from the five independent runs performed to generate uncertainty bands for this plot, every point on this plot is from the same underlying network, just with different masks. See Figure S6 for performance on all four networks.

allows us to produce masks with the amounts of pruning (percentages of zeros) ranging from 7% to 89%. Further details about the training can be seen in Section S6.

## 5.2 Dynamic Weight Rescaling

One beneficial trick in Supermask training is to dynamically rescale the values of weights based on the sparsity of the network in the current training iteration. For each training iteration and for each layer, we multiply the underlying weights by the ratio of the total number of weights in the layer over the number of ones in the corresponding mask. Dynamic rescaling leads to significant improvements in the performance of the masked networks, which is illustrated in Table 1.

Table 1 summarizes the best test accuracy obtained through different treatments. The result shows striking improvement of learned Supermasks over heuristic based ones. Learned Supermasks result in performance close to training the full network, which suggests that a network upon initialization already contains powerful subnetworks that work well without training.

## 6 Conclusion

In this paper, we have studied how three components to LT-style network pruning—mask criterion, treatment of kept weights during retraining (mask-1 action), and treatment of pruned weights during retraining (mask-0 action)—come together to produce sparse and performant subnetworks. We proposed the hypothesis that networks work well when pruned weights are set close to their final values. Building on this hypothesis, we introduced alternative freezing schemes and other mask

Table 1: Test accuracy of the best Supermasks with various initialization treatments. Values shown are the max over any prune percentage and averaged over four or more runs. The first two columns show untrained networks with heuristic-based masks, where "init" stands for the initial, untrained weights, and "S.C." is the signed constant approach, which replaces each random initial weight with its sign as described in Section 3. The next two columns show results for untrained weights overlaid with learned masks; and the two after add the Dynamic Weight Rescaling (DWR) approach. The final column shows the performance of networks with weights trained directly using gradient descent. Bold numbers show the performance of the best Supermask variation.

| Network | mask $\odot$ init | mask $\odot$ S.C. | learned mask $\odot$ init | learned mask $\odot$ S.C. | DWR learned mask $\odot$ init | DWR learned mask $\odot$ S.C. | trained weights |
|---|---|---|---|---|---|---|---|
| MNIST FC | 79.3 | 86.3 | 95.3 | 96.4 | 97.8 | **98.0** | 97.7 |
| CIFAR Conv2 | 22.3 | 37.4 | 64.4 | **66.3** | 65.0 | 66.0 | 69.2 |
| CIFAR Conv4 | 23. | 39.7 | 65.4 | 66.2 | 71.7 | **72.5** | 75.4 |
| CIFAR Conv6 | 24.0 | 41.0 | 65.3 | 65.4 | 76.3 | **76.5** | 78.3 |

criteria that meet or exceed current approaches by respecting this basic rule. We also showed that the only element of the original initialization that is crucial to the performance of LT networks is the sign, not the relative magnitude of the weights. Finally, we demonstrated that the masking procedure can be thought of as a training operation, and consequently we uncovered the existence of Supermasks, which can produce partially working networks without training.

## Acknowledgments

The authors would like to acknowledge Jonathan Frankle, Joel Lehman, Zoubin Ghahramani, Sam Greydanus, Kevin Guo, and members of the Deep Collective research group at Uber AI for combinations of helpful discussion, ideas, feedback on experiments, and comments on early drafts of this work.

## Footnotes

[1]The early stopping criterion we employ in this paper is the iteration of minimum validation loss.

[2]We run a t-test for each pruning percentage based on a sample of 5 independent runs for each mask criteria.

[3]Figure S3 illustrates why the `large_final` criterion biases weights that were moving toward zero during training toward zero in the mask, effectively pushing them further in the direction they were headed.

[4]Additional control variants of this experiment can be seen in Supplementary Information Section S3.

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
