[Supplementary Material]

# Supplementary Information for: Deconstructing Lottery Tickets: Zeros, Signs, and the Supermask

## S1  Architectures and training hyperparameters

Table S1 contains the architectures used in this study, together with relevant training hyperparameters, based off of experiments in [5].

**Additional details**    To compare with results in [5], we used the same training hyperparameters and did not do any additional tuning. Training/test splits were given in both MNIST and CIFAR-10; validation was split randomly from the training set with (55000, 5000) train and val for MNIST and (45000, 5000) for CIFAR.

Our experiments required more computation than regular training procedures, as networks were trained up to 24 times with iterative pruning. We used single GPUs for each experiment (NVIDIA GeForce GTX 1080 Ti) and parallelized by running multiple experiments on multiple GPUs.

Table S1:  The architectures used in this paper.  Table reproduced and modified from [5].  Conv networks use 3x3 convolutional layers with max pooling followed by fully connected layers. FC layer sizes are from [12]. Initializations are Glorot Normal [7] and activations are ReLu.

| Network | MNIST FC | CIFAR-10 Conv2 | CIFAR-10 Conv4 | CIFAR-10 Conv6 |
|---|---|---|---|---|
| *Convolutional Layers* | | 64, 64, pool | 64, 64, pool 128, 128, pool | 64, 64, pool 128, 128, pool 256, 256, pool |
| *FC Layers* None | 300, 100, 10 | 256, 256, 10 | 256, 256, 10 | 256, 256, 10 |
| *All/Conv Weights* | 266K | 4.3M / 38K | 2.4M / 260K | 2.3M / 1.1M |
| *Iterations/Batch* | 50K / 60 | 20K / 60 | 25K / 60 | 30K / 60 |
| *Optimizer* | Adam 1.2e-3 | Adam 2e-4 | Adam 3e-4 | Adam 3e-4 |
| *Pruning Rates* | fc20% | conv10% fc20% | conv10% fc20% | conv15% fc20% |

## S2  Further mask criteria details

Figure S1 shows the convergence speed and performance of all mask critera for FC on MNIST and Conv2, 4, 6 on CIFAR-10.

## S3  Further mask-0 action details

In this section, we discuss additional control variants of the mask-0 action experiments shown in Figure 5. In particular, we want to know if the improvement in performance from setting only weights that move towards zero to zero is the result of this particular selection of zero weights, rather than other quirks in the treatment. We run two control cases. In the first control experiment, we randomly freeze a subset of pruned weights to zero, with the number of zero-ed weights equal to the number of

zero-ed weights in "set pruned to init or 0". In the second control experiment, we set weights that *moved away from zero* to zero, reversing the proposed treatment. As shown in Figure S2, both control experiments perform significantly worse than the proposed treatment ("set pruned to init or 0"), with the reversed treatment performing worse than both randomly freezing weights to zero and freezing all pruned weights at initial value. These results are consistent with our hypothesis.

## S4   Further mask-1 action details

Figure S4 shows the convergence speed and performance of various reinitialization methods for FC on MNIST and Conv2, 4, 6 on CIFAR-10.

## S5   Further heuristic Supermask details

We show the performance of trained LT networks with mask criteria large_final_same_sign and large_final_diff_sign in Figure S5. We see that despite having better test accuracy at iteration 0, large_final_same_sign slightly underperforms large_final when the underlying network is fully trained. This may be due to the fact that the large_final_same_sign mask criterion randomly select weights that changed signs to keep when there are not enough same sign weights.

## S6   Further training details for learning Supermasks

We train the networks with mask $m$ for each layer (and all regular kernels and biases frozen) with SGD, 0.9 momentum. The {FC, Conv2, Conv4, Conv6} networks respectively had {100, 100, 50, 20} for learning rates and trained for {2000, 2000, 1000, 800} iterations. These hyperparameters may seem absurd, but a network of masks is quite different and cannot train well with typical learning rates. Conv4 and Conv6 showed significant overfitting, thus we used early stopping as we are unable to use standard regularizing techniques. For evaluation, we also use Bernoulli sampling, but average the accuracies over 10 independent samples.

For adjusting the amount pruned, we initialized $m$ in every layer to be the same constant, which ranged from -5 to 5. In the future it may be worth trying different initializations of $m$ for each layer for more granular control over per-layer pruning rates. A different method to try would be to add an L1 loss to influence layers to go toward certain values, which may alleviate the cold start problems of some networks not learning anything due to mask values starting too low (effectively having the entire network start at zero).

Figure S1: Performance of different mask criteria for four networks at various pruning rates. We show early stopping iteration on the left and test accuracy on the right. Each line is a different mask criteria, with bands around magnitude_increase, large_final, movement, and random depicting the min and max over 5 runs. Stars represent points where large_final or magnitude_increase are significantly above the other at a $p < 0.05$ level. large_final and magnitude_increase show the best convergence speed and accuracy, with magnitude_increase having slightly higher accuracy in Conv2 and Conv4. As expected, criteria using small weight values consistently perform worse than random.

Figure S2: Performance of network pruning using different treatments of pruned weights (mask-0 actions). Horizontal black lines represent the performance of training the original, full network, averaged over five runs. Solid blue lines represent the original LT algorithm, which freezes pruned weights at zero. Dotted blue lines freeze pruned weights at their initial values. Grey lines show the new proposed 0-action—set to zero if they decreased in magnitude by the end of training, otherwise set to their initialization values. Two variants are shown: 1) new treatment applied to only pruned weights (dashdotted grey lines); 2) new treatment applied to all weights (dashed grey lines). Two additional control experiments are shown: 1) randomly freeze a number of pruned weights to zero, with the number equal to the number of zero-ed weights in the new proposed 0-action ("set pruned to init or 0"); 2) reverse the proposed 0-action by setting to zero weights that *increased* in magnitude by the end of training.

Figure S3: The motion selectivity of the large_final mask criterion. Although large_final only selects for large final weights, not directly for weight motion, it biases towards pruning weights that *decreased* in magnitude, since those weights are more likely to have small final magnitudes. As shown by the two paired sets of example weights, those two weights that increased during training (two solid up arrows) are reset to their initial value (dotted arrows) after pruning, resulting in no net motion. In contrast, those two weights that decrease in value during training (two solid down arrows) are set to zero (dotted lines) during pruning, moving them preferentially in the direction they moved during training. This explains why Supermasks may be created by such a simple criterion as large_final and motivates the more specifically designed, higher performing large_final_same_sign criterion (depicted in Figure 6).

Figure S4: The effects of various 1-actions for the four networks and various pruning rates. Dotted lines represent the three described methods, and solid lines are those three except with each weight having the same sign as its original initialization. Shaded bands around notable runs depict the min and max over 5 runs. Stars represent points where "rewind (large_final)" or "constant, init sign" is significantly above the other at a $p < 0.05$ level, showing no difference in performance between the two. We also include the original rewinding method and random pruning as baselines. "Reshuffle, init sign" and "constant, init sign" perform similarly to the "rewind" baseline.

Figure S5: Test accuracy at early stopping iteration of additional mask criteria (large_final, large_final_same_sign, large_final_diff_sign) for four networks at various pruning rates.

Figure S6: Comparision of Supermask performances in terms of test accuracy on MNIST and CIFAR-10 classification tasks. Subfigures are across various network structures (from top row to bottom: FC on MNIST, Conv2 on CIFAR-10, Conv4 on CIFAR-10, Conv6 on CIFAR-10), as well as 1-action treatments (left: weights are frozen at their original initialization, right: weights are frozen at a signed constant). No training is performed in any network. Weights are frozen at either initialization or constant and various masks are applied. Within heuristic based Supermasks (excluding learned_mask), the large_final_same_sign mask creates the highest performing Supermask by a wide margin. Note that aside from the five independent runs performed to generate uncertainty bands for this plot, every data point on the plot is the same underlying network, just with different masks.