[Reviews · NeurIPS 2019]

Reviewer 1



After reading the feedback, I am still not fully convinced that the results are significant, but I think the feedback makes sense. Therefore I've increased my score to 5 (weak rej). Here is the reason. I think this is an interesting paper, and the observations are good to know. However, maybe because I am from theory community, I feel the results are not that significant for the following reasons. 1. If an empirical paper has significant results, I usually believe that they should provide better empirical performance, e.g., better test accuracy, running time, memory requirement, etc. This paper does not provide a better algorithm. The best mask they find has similar performance as the existing one. 2. Since this paper does not have better empirical performance, I tend to hope that it will give us better understanding of lottery hypothesis. However, (a) First, all the experiments are run on small datasets, like MNIST/CIFAR, and not on large datasets, so it is not clear whether the claimed observation is universal. (b) Secondly, it is not clear whether the presented observations are useful. For example, the authors show that LT is also valid on "magnitude increase" mask, and keeping the sign of initial value is more important and keeping the value. I agree those are interesting observations, but I don't know why these observations are important or significant. Is it for better pruning algorithm? But the authors say the performance is similar. Is it for understanding the training process of neural networks? But the authors did not mention any explicit connections, and I couldn't think of any as well. If LT becomes a universal algorithm that will be applied everywhere, then I think the exploration made in this paper might be valuable. But currently this is still not the case, as far as I know. Therefore, personally I would not be willing to increase my score to weak accept/accept for this paper. ---------------- Originality: this paper is doing ablation study on the previous lottery ticket paper. Therefore, the originality is limited. Quality: this submission is purely empirical, and the authors use the plots to show their claims. The experiments are only on CIFAR-10 and MNIST, so not very convincing. Clarity: I think the clarity can be further improved. There are so many different kinds of masks, and it seems to me that most of them do not work well. If that's the case, maybe the authors should consider remove some irrelevant curves, to make the plots easier to read. Moreover, it is hard for me to get the main message from the text. Significance: I think the results are interesting, but I am not sure whether they are significant. For the first result, the outcomes of "magnitude increase" and "large final" look similar to me, and it is not clear what the authors want to show here. For the second result, keeping the sign of the original initial value looks very interesting, however I do not know what the indication of this observation is. For the final result, masking is training is a nice claim, but again, I do not know what is the message that the author wants to deliver. I feel this paper still needs some work on showing why the results are important.

Reviewer 2



After reading the author's reply, my review is unchanged; I still support acceptance. (I'm not sure including an image which contains a lot of text at half the font size, plus new plots, is appropriate to include in the author's reply, but I would let it slide here.) The main reasons for my unchanged score: 1. Even though this work doesn't push the state of the art on a new task, it presents useful and interesting information, which really is the goal of research. 2. While I don't think these exact results will hold on more difficult datasets, or those in other fields (like text classification), I do think similar trends would exist. 3. While the authors don't speculate on how their results could be used, their experiments were conducted well enough that I believe they will be useful to future researchers. For example, I'd like to see a paper following up on this which does an indepth analysis of the types of structures learned using such masks, which would lead to better neural model structures. Similarly, I'd like to see a paper analyzing the initializations, and leading to better initialization schemes. These both seem beyond the scope of this work, but would build directly upon it. My original review is below. =================================== This work is original, as I don't believe these mask criteria have been analyzed before. That said, this work doesn't draw connections to the rich history of pruning methods, even though many are similar (perhaps that should be another paper altogether, though). The quality of the work is fairly high, though there are a few missing experiments (like the trained performance of the supermasks). I believe future work will build upon the results in this paper, and that it is significant. It's good the authors include detailed descriptions of the models used in the appendix, instead of just relying on referencing previous work.

Reviewer 3



# Response to rebuttal I would like to thank the authors for their rebuttal. Overall, my opinion about the manuscript remains unchanged. Despite some drawbacks, such as the lack of results using more challenging datasets and/or more complex architectures, I believe the paper presents a well-executed empirical analysis that addresses some questions about the LTH, as well as raises some new ones, making it interesting for future work. Therefore I will keep my previous score, supporting acceptance. # Summary In this paper, the authors present an empirical study to investigate the recently proposed “Lottery Ticket Hypothesis” by means of ablation experiments. In brief, “Lottery Tickets” (LTs) are sparse subnetworks which can be (re)-trained in isolation and yet achieve comparable or even better performance than the original, unpruned network. Prior work identified these LTs by first training the full network and then 1) selecting the (100-p)% weights with smallest absolute magnitude after training as weights to be pruned; 2) resetting the p% weights to be kept to their initial value; and 3) setting all (100-p)% pruned weights to zero and keeping them frozen to this value when (re)-training the subnetwork. In particular, prior work found step 2) to be crucial; if the p% weights to be kept are not reset to their initial value but rather randomly re-initialized, the (re)-trained LT is no longer able to perform comparably to the full network. All in all, prior work presented compelling evidence backing the “Lottery Ticket Hypothesis”, yet many aspects of these findings remained poorly characterized. In this manuscript, a comprehensive set of ablation studies were performed to empirically explore additional degrees of freedom for each of the three steps above. Specifically: 1) Alternative criteria to select the set of weights to be pruned were considered, including initial magnitude, a combination of initial and final magnitude, magnitude increase and absolute difference between initial and final weight values. Additionally, “sanity-check” control criteria were also included, such as random selection and “inverted” versions of some of the aforementioned approaches. 2) Alternative ways to reset the value of the weights to be kept were considered, including random re-initialization, permuting the initial values within each layer and initializing the weights to constant values with a random sign. Additionally, alternative versions of each criterion guaranteeing that the sign of the resetted weights agrees with the sign of the initial weights were also included. 3) Alternative ways to select the values at which to freeze the pruned weights were considered, including freezing them at their initial value, setting them to either zero or their initial value depending on whether they moved towards zero or not after training the full network and “sanity-check” controls such as randomly setting them to either zero or their initial value. The results of these ablation studies provided clear additional insight on the “Lottery Ticket Hypothesis”. In particular: 1) the phenomenon is not exclusive to the original criterion used to select the set of weights to be pruned; 2) resetting the kept weights to their initial value does not appear to be strictly necessary as reported by prior work, but keeping the signs identical seems to be crucial; and 3) pruned weights can be set to either zero or their initial value when the decision is informed by the dynamics of training in the full network, suggesting that masking pruned weights of small magnitude after training might work partly because it is somewhat consistent with the training dynamics. Finally, the authors also introduced the concept of “Supermasks” by observing that LTs have better-than-random accuracy even before training the subnetwork, showing that despite being computed in crude ways, these masks can encode substantial information about the (supervised) learning task at hand. # High-Level Assessment The recent discovery of the “Lottery Ticket Hypothesis” has led to many unanswered questions of high relevance to the field, some of which this paper explored with a thoughtful and well-executed set of ablation experiments. Despite still largely lacking a theoretical characterization of the phenomenon, I believe the empirical results reported in this manuscript would be of interest to the community and might pave the way to impactful future work. The paper is very well-written, being generally a pleasure to read. Moreover, all aspects of the experimental setup are clearly detailed and seem reproducible, which is of particular importance in a paper with empirical results as its main contribution. Because of all the aforementioned reasons, I am in favour of accepting the manuscript for publication. # Other Comments 1. The main conclusion of Section 3 seems to be that as long as the weights to be kept are resetted to a value with the same sign as they had initially, (re)-training the LT will be successful. However, as shown in Figure 4, while this trend indeed seems hold generally, it can also be noticed that randomly re-initializing the weights can sometimes perform poorly even when the signs are respected (e.g. Conv4 and Conv6), as originally reported by Frankle & Carbin. Do the authors have any conjecture about why this might be the case? Is there something else essential for the “Mask-1 action" other than keeping the sign? 2. Regarding the optimization of “Supermasks” described in Section 5.1: 2.1 Given that the masks are stochastic, how are the LTs used at testing time? Are they substituted by their expectation, or is a MC estimate used instead (if so, with how many samples)? 2.2 Has there been any attempt to initialize the “Supermasks” prior to SGD-based optimization by, for example, making the initial value of m proportional to the scores of the “large_final_diff_sign” criterion? 2.3 Do the initial, frozen weights play a crucial role in these experiments? Can good performance be achieved if these weights were, for example, set to have constant magnitude with random signs? 3. Perhaps a natural extension of the different masking criteria introduced in Section 2 would be to consider the entire optimization trajectory instead of only the endpoints. Have the authors considered in any preliminary experiment to, for example, substitute the initial value of the weights in their criteria by the value of the weights after a few steps of training? # Typos Line 183: treaments -> treatments Lines 209 - 210: there might be a missing word

[Author Response · NeurIPS 2019]

We thank reviewers sincerely for their detailed feedback. It has enabled us to improve and clarify several sections.

**1. The results are not tested on ImageNet / ResNet [R1, R2, R3].** We agree it would be great to scale these
experiments to deeper models (ResNet) and larger datasets (ImageNet) and see how well they generalize. As found by
Frankle et al in "Stabilizing..." (2019), not all Lottery Ticket (LT) results generalized to ImageNet-sized models. We
suspect that some of our results may similarly not generalize (such as mask-1 actions) but some will (such as Supermasks
and mask-0 actions because our hypothesis on masking as training is independent of the original initialization of the
weights). Due to time and computational constraints we leave this for future work, but have added a note in our paper
discussing the lack of generalization results. *[See Update a below]*

**2. Significance of results are unclear; mask criteria curves unclear [R1].** Some of the results were indeed poorly
described; we have updated several descriptions and the plot captions and think the clarity has been improved
significantly *[Updates b, e, f]*. Summary: the significance of "magnitude increase" results are two-fold: it shows the LT
phenomenon is not exclusive to the large final criterion, and the results are consistent with our hypothesis of masking as
training. The mask-1 "significance of the sign" results show, in contrast to the LT paper, that the basin of attraction
for a lottery ticket network is actually quite large: anywhere in the correct weight quadrant optimizes well; optimizers
encounter difficulty crossing the zero barrier between signs. Finally, "masking as training" proposes an explanation for
many of our results (and is predictive of performance of both mask criteria and alternate mask-0 treatments). We have
also made the plots easier to interpret by simplifying the confidence bands and highlighting the main takeaways in the
caption *[Updates d]*.

**3. Missing results on trained performance of additional Supermasks [R2].** This is a great suggestion. We have
run these experiments and included them in SI *[Update g]*. We found that large_final_same_sign actually slightly
underperforms large_final when trained in the iterative pruning LT paradigm. This is likely because we break 0 ties
randomly when a mask criterion does not contain enough 1's, and selecting only weights with the same sign entails the
need for much random selection.

**4. Questionable use of statistical significance [R2].** Iterative pruning does indeed cause correlation across different
pruning percentages. However, we never report p-values aggregated in this way: all statistical tests are performed at a
single pruning percentage, aggregating only across five independent runs. We've clarified in the text *[Update h]*.

**5. Mask-0 experiments and Fig 5 poorly motivated [R2].** We've removed the unnecessary straw man argument and
state the real motivation more simply: to test the hypothesis that the value of frozen weights affects training *[Update c]*.

**6. Further analysis of masks themselves + principled understanding of lottery tickets [R2, R3].** We agree that
it would be a valuable research direction to better understand and provide a theoretical foundation for what makes
good masks, as well as their connection to better initializations. We believe our results represent a significant though
not consummate step toward better understanding. Relating to the **question on the performance of signed_reinit**
**[R3]**, our results hint that using a bimodal distribution to initialize weights may be more beneficial than a Gaussian
distribution. This may explain why signed_reinit underperforms signed_reshuffle and signed_constant,
both of which are initialized using a bimodal distribution.

**7. Clarifications on Supermask training [R2, R3].** The Bernoulli is sampled each forward pass, so during training
each mini-batch entails a fresh sample of the mask. At test time, we report the average accuracy over 10 random samples
of the mask (aggregating accuracy of the single-sample models, not ensembling predictions). As **[R3]** mentions this is
akin to learning a per-weight dropout probability, but we should be careful not to interpret this as learned regularization
but as the entire learning process itself. **[R3 point 2.3]** We respond to two possible interpretations of your suggestion
here: (1) Supermasks are indeed trainable from initial weights that are const magnitude, random sign (Fig 7, right). (2)
If signs are randomized *after* training a Supermask, performance degrades to chance (experiment run but not shown).

**8. Initialize the learned Supermasks based on a well-performing heuristic mask criteria [R3].** This is a great idea
and would likely work, though for now, because it works and is simple, we have trained only from scratch.

**9. Mask using other points on optimization trajectory instead of endpoints [R3].** Great idea. We have not yet
explored computing masks using (intermediate, final) weights as you suggest instead of (initial, final) weights. We did
try masking using (initial, min-val-err) weights, but performance was not as good.



[Meta-Review · NeurIPS 2019]

Thanks to the authors for providing a rebuttal, it really helped drive the subsequent discussion between myself and the reviewers. Ultimately, we could not reach an overwhelming consensus amongst the reviewers. But I found the arguments presented by the positive reviews to be compelling enough that I will argue for acceptance of this work. The main contribution being a thorough empirical analysis of the LTH paper, which is an exciting result that would definitely benefit from more understanding in the community. I think the empirical kind of understanding is valuable so despite the lack of theory so I will be recommending this paper be accepted.